# Neutrophil Immunomodulatory Activity of (−)-Borneol, a Major Component of Essential Oils Extracted from *Grindelia squarrosa*

**DOI:** 10.3390/molecules27154897

**Published:** 2022-07-31

**Authors:** Igor A. Schepetkin, Gulmira Özek, Temel Özek, Liliya N. Kirpotina, Andrei I. Khlebnikov, Mark T. Quinn

**Affiliations:** 1Department of Microbiology and Cell Biology, Montana State University, Bozeman, MT 59717, USA; igor@montana.edu (I.A.S.); liliya@montana.edu (L.N.K.); 2Department of Pharmacognosy, Faculty of Pharmacy, Anadolu University, Eskisehir 26470, Turkey; gulmiraozek@gmail.com (G.Ö.); temelozek@gmail.com (T.Ö.); 3Kizhner Research Center, Tomsk Polytechnic University, 634050 Tomsk, Russia; aikhl@chem.org.ru

**Keywords:** borneol, calcium influx, chemotaxis, essential oils, *Grindelia squarrosa*, monoterpene, neutrophil

## Abstract

*Grindelia squarrosa* (Pursh) Dunal is used in traditional medicine for treating various diseases; however, little is known about the immunomodulatory activity of essential oils from this plant. Thus, we isolated essential oils from the flowers (GEO_Fl_) and leaves (GEO_Lv_) of *G. squarrosa* and evaluated the chemical composition and innate immunomodulatory activity of these essential oils. Compositional analysis of these essential oils revealed that the main components were α-pinene (24.7 and 23.2% in GEO_Fl_ and GEO_Lv_, respectively), limonene (10.0 and 14.7%), borneol (23.4 and 16.6%), *p*-cymen-8-ol (6.1 and 5.8%), β-pinene (4.0 and 3.8%), bornyl acetate (3.0 and 5.1%), *trans*-pinocarveol (4.2 and 3.7%), spathulenol (3.0 and 2.0%), myrtenol (2.5 and 1.7%), and terpinolene (1.7 and 2.0%). Enantiomer analysis showed that α-pinene, β-pinene, and borneol were present primarily as (−)-enantiomers (100% enantiomeric excess (ee) for (−)-α-pinene and (−)-borneol in both GEO_Fl_ and GEO_Lv_; 82 and 78% ee for (−)-β-pinene in GEO_Fl_ and GEO_Lv_), while limonene was present primarily as the (+)-enantiomer (94 and 96 ee in GEO_Fl_ and GEO_Lv_). *Grindelia* essential oils activated human neutrophils, resulting in increased [Ca^2+^]_i_ (EC_50_ = 22.3 µg/mL for GEO_Fl_ and 19.4 µg/mL for GEO_Lv_). In addition, one of the major enantiomeric components, (−)-borneol, activated human neutrophil [Ca^2+^]_i_ (EC_50_ = 28.7 ± 2.6), whereas (+)-borneol was inactive. Since these treatments activated neutrophils, we also evaluated if they were able to down-regulate neutrophil responses to subsequent agonist activation and found that treatment with *Grindelia* essential oils inhibited activation of these cells by the *N*-formyl peptide receptor 1 (FPR1) agonist *f*MLF and the FPR2 agonist WKYMVM. Likewise, (−)-borneol inhibited FPR-agonist-induced Ca^2+^ influx in neutrophils. *Grindelia* leaf and flower essential oils, as well as (−)-borneol, also inhibited *f*MLF-induced chemotaxis of human neutrophils (IC_50_ = 4.1 ± 0.8 µg/mL, 5.0 ± 1.6 µg/mL, and 5.8 ± 1.4 µM, respectively). Thus, we identified (−)-borneol as a novel modulator of human neutrophil function.

## 1. Introduction

*Grindelia* was established by Willdenow in 1807 and is recognized as one of the most distinct and homogeneous genera in the Compositae family [1]. The genus *Grindelia* Willd. comprises about 68 species [2], and the following four species have been used medicinally: *Grindelia robusta* Nutt. (syn. *G. camporum* Greene, California gumdrop), *G. hirsutula* Hook. & Arn. (syn. *G. humilis* Hook. & Arn., Hairy Gumweed), *G. lanceolata* Nutt. (Narrow-leaved gumdrop), and *G. squarrosa* (Pursh) Dunal (gumweed, bulky rubber weed) [3]. For example, grindelic acid isolated from *G. squarrosa* was reported to have anti-inflammatory activity [4], and previous pharmacological studies emphasized the possibility of the application of *G. robusta* and *G. squarrosa* in diseases of the upper respiratory tract [5]. Likewise, Gumweed herb (*G. camporum*) was indicated for cough and bronchitis [6]. Due to its antispasmodic, expectorant, and hypotensive effects, *G. squarrosa* was suggested for the treatment of asthmatic and bronchial conditions, especially when these are associated with rapid heartbeat and nervous responses [7].

*Grindelia squarrosa* (Pursh) Dunal is a biennial or a short-lived perennial plant that is native to North America [1]. Its range comprises the central prairie and plains regions of southern Minnesota and South Dakota, Wyoming, Iowa, Nebraska, Kansas, and Texas [8]. American Indians have used *Grindelia* flowers and leaves for treating colds, coughs, bronchitis, and asthma (Crow, Flathead, Paiute, Shoshoni, and Ute). In addition, the Cheyenne people used a decoction of this plant as a disinfectant to wash sores and other skin lesions. Likewise, a decoction of *Grindelia* roots was used for liver problems by the Blackfoot tribe [9].

*Grindelia* species contain valuable secondary metabolites that have pharmacological importance, including essential oils [10,11,12,13,14,15,16], labdane type diterpenes [17,18], triterpene saponins [19], flavonoids [19,20], tannins [21], and phenolic acids [22]. Previous analyses of the biological activity of extracts from *Grindelia* species resulted in the identification of cytotoxic, anti-inflammatory [4,19,23,24], antimicrobial [12], and free radical scavenging [25] properties. 

Essential oils from various plant species have been reported to exhibit anticancer, anti-inflammatory, and immunomodulatory effects (reviewed in [26,27,28]). The most important immunopharmacological mechanisms of essential oils reported were the regulation of cytokine production, inhibition of reactive oxygen species (ROS) production, and inactivation of eosinophil migration [27]. In addition, we recently found that essential oils from *Artemisia kotuchovii* Kupr., *Ferula akitschkensis* B.Fedtsch. ex Koso-Pol., *Ferula iliensis* Krasn. ex Korovin, *Hypericum perforatum* L., *Rhododendron albiflorum* Hook., and *Juniperus* and *Artemisia* spp. can modulate human neutrophil functions [29,30,31,32,33,34]. Moreover, essential oils and some of their component compounds (monoterpenes, sesquiterpenes) have been reported to modulate humoral and cellular immune responses [35,36,37,38].

In the present study, we isolated essential oils from the leaves and flowers of *G. squarrosa* collected in Montana and analyzed their chemical compositions and innate immunomodulatory activities. We showed that essential oils isolated from *G. squarrosa* leaves and flowers potently inhibited intracellular Ca^2+^ mobilization [Ca^2+^]_i_ in human neutrophils. Furthermore, we demonstrated that (−)-borneol, which is present at high levels in *G. squarrosa* essential oils, inhibited human neutrophil functional responses and that this compound is likely one of the main active components in these essential oils. Given the critical role of neutrophils in inflammation, our data support the possibility that (−)-borneol could be considered in the development of anti-inflammatory agents.

## 2. Materials and Methods

### 2.1. Plant Material

Plant material was collected from wild plants in August 2019 along the Madison River, approximately 10 miles east of Norris, MT, USA (45.611323° N, 111.564253° E). Flowers and leaves were air-dried for 7–10 days at room temperature away from direct sunlight before hydrodistillation. Botanical identification of the plant material was performed by botanist Robyn A. Klein from Montana State University, Bozeman, MT, USA. 

### 2.2. Materials

Dimethyl sulfoxide (DMSO), *N*-formyl-Met-Leu-Phe (*f*MLF), Trp-Lys-Tyr-Val-Met (WKYMVM), and Histopaque 1077 were purchased from Sigma-Aldrich Chemical Co. (St. Louis, MO, USA). (−)-Borneol and (+)-borneol were from Cayman Chemical Company (Ann Arbor, MI, USA). *n*-Hexane was purchased from Merck (Darmstadt, Germany). Fluo-4AM was purchased from Invitrogen (Carlsbad, CA, USA). Roswell Park Memorial Institute (RPMI) 1640 medium was purchased from HyClone Laboratories (Logan, UT, USA). Fetal calf serum and fetal bovine serum were purchased from ATCC (Manassas, VA, USA). Hanks’ balanced salt solution (HBSS; 0.137 M NaCl, 5.4 mM KCl, 0.25 mM Na_2_HPO_4_, 0.44 mM KH_2_PO_4_, 4.2 mM NaHCO_3_, 5.56 mM glucose, and 10 mM HEPES, pH 7.4) was purchased from Life Technologies (Grand Island, NY, USA). HBSS without Ca^2+^ and Mg^2+^ was designated as HBSS^–^; HBSS containing 1.3 mM CaCl_2_ and 1.0 mM MgSO_4_ was designated as HBSS^+^. 

### 2.3. Essential Oil Extraction

Essential oils were extracted by hydrodistillation of air-dried plant material (leaves, flowers) using a Clevenger-type apparatus, as previously described [34]. We used conditions accepted by the European Pharmacopoeia (European Directorate for the Quality of Medicines, Council of Europe, Strasbourg, France, 2014) to avoid artifacts. Yields of the essential oils were calculated based on the amount of air-dried plant material used. Stock solutions of the essential oils were prepared in DMSO (10 mg/mL) for biological evaluation and in *n*-hexane (10% *w*/*v*) for gas chromatographic analysis.

### 2.4. Gas Chromatography–Flame Ionization Detector (GC-FID) and Gas Chromatography–Mass Spectrometry (GC-MS) Analysis

GC-MS analysis was performed with an Agilent 5975 GC-MSD system (Agilent Technologies, Santa Clara, CA, USA), as reported previously [39]. An Agilent Innowax FSC column (60 m × 0.25 mm, 0.25 μm film thickness) was used with He as the carrier gas (0.8 mL/min). The GC oven temperature was kept at 60 °C for 10 min, increased to 220 °C at a rate of 4 °C/min, kept constant at 220 °C for 10 min and then increased to 240 °C at a rate of 1 °C/min. The split ratio was adjusted to 40:1, and the injector temperature was 250 °C. MS spectra were monitored at 70 eV with a mass range of 35 to 450 *m*/*z*. GC analysis was performed on an Agilent 6890N GC system. To obtain the same elution order as with GC-MS, the line was split for FID and MS detectors, and a single injection was performed using the same column and appropriate operational conditions. FID temperature was 300 °C. The essential oil components were identified by co-injection with standards (whenever possible), which were purchased commercially or isolated from natural sources. In addition, compound identities were confirmed by comparison of their mass spectra with those in the Wiley GC/MS Library (Wiley, NY, USA), MassFinder software 4.0 (Dr. Hochmuth Scientific Consulting, Hamburg, Germany), Adams Library, and NIST Library. Confirmation was also achieved using the in-house “Başer Library of Essential Oil Constituents” database obtained from chromatographic runs of pure compounds performed with the same equipment and conditions. A C_8_–C_40_ *n*-alkane standard solution (Fluka, Buchs, Switzerland) was used to spike the samples for the determination of relative retention indices (RRI). Relative percentage amounts of the separated compounds were calculated from the FID chromatograms.

### 2.5. Analysis of Enantiomer Composition on Chiral Columns

Distributions of enantiomeric compounds in *G. squarrosa* essential oils were analyzed by using two different chiral columns, which were appropriate for each enantiomer, and an Agilent 5973 Network Mass Selective Detector on a 6890N GC system that also had an FID detector (Agilent Technologies, Santa Clara, CA, USA) (see further GC system details above). The chiral columns were Rt-βDEXse (2,3-di-*O*-ethyl-6-*O*-tert-butyl dimethylsilyl-β-cyclodextrin added into 14% cyanopropylphenyl/86% dimethyl polysiloxane, 30 m × 0.32 mm ID, 0.25 μm film thickness, USA) and Lipodex G (6-methyl-2,3-pentyl-γ-cyclodextrin added into 60% polysiloxane, 25 m × 0.25 mm ID, 0.125 µm film thickness, Germany). For separation of α-pinene, the Lipodex G chiral column was used, while separation of borneol, camphor, and limonene was performed on the Rt-βDEXse column. Samples were injected (10% prepared in hexane) with a 10:1 split ratio. Injection port and detector temperatures were 250 °C. Detailed analysis parameters are provided in Appendix A.

### 2.6. Isolation of Human Neutrophils

For isolation of human neutrophils, blood was collected from healthy donors in accordance with a protocol approved by the Institutional Review Board at Montana State University (protocol #MQ041017). Neutrophils were purified from the blood using dextran sedimentation, followed by Histopaque 1077 gradient separation and hypotonic lysis of red blood cells, as described previously [40]. Isolated neutrophils were washed twice and resuspended in HBSS^–^. Neutrophil preparations were routinely >95% pure, as determined by light microscopy, and >98% viable, as determined by trypan blue exclusion. Neutrophils were obtained from multiple different donors; however, the cells from different donors were never pooled during experiments.

### 2.7. Cell Culture

Human THP-1 monocytic cells obtained from ATCC (Manassas, VA, USA) were cultured in RPMI 1640 medium (Mediatech Inc., Herndon, VA, USA) supplemented with 10% (*v*/*v*) FBS, 100 μg/mL streptomycin, and 100 U/mL penicillin.

### 2.8. Ca^2+^ Mobilization Assay

Changes in intracellular Ca^2+^ concentrations ([Ca^2+^]_i_) were measured with a FlexStation 3 scanning fluorometer (Molecular Devices, Sunnyvale, CA, USA). Briefly, human neutrophils were suspended in HBSS^-^, loaded with Fluo-4AM at a final concentration of 1.25 μg/mL, and incubated for 30 min in the dark at 37 °C. After dye loading, the cells were washed with HBSS^-^, resuspended in HBSS^+^, separated into aliquots, and loaded into the wells of flat-bottom, half-area-well black microtiter plates (2 × 10^5^ cells/well). To assess the direct effects of a test compound or pure essential oils on Ca^2+^ influx, the compound/oil was added to the wells (final concentration of DMSO was 1%), and changes in fluorescence were monitored (λ_ex_ = 485 nm, λ_em_ = 538 nm) every 5 s for 240 s at room temperature after addition of the test compound. To evaluate inhibitory effects of the compounds on FPR1/FPR2-dependent Ca^2+^ influx, the compound/oil was added to the wells (final concentration of DMSO was 1%) with human neutrophils. The samples were preincubated for 10 min, followed by addition of 5 nM *f*MLF or 5 nM WKYMVM. The maximum change in fluorescence, expressed in arbitrary units over baseline, was used to determine the agonist response. Responses were normalized to the response induced by 5 nM *f*MLF or 5 nM WKYMVM, which were assigned as 100%. Curve fitting (at least five or six points) and calculation of median effective concentration values (EC_50_ or IC_50_) were performed by nonlinear regression analysis of the dose–response curves generated using Prism 9 (GraphPad Software, Inc., San Diego, CA, USA).

### 2.9. Chemotaxis Assay

Human neutrophils were resuspended in HBSS^+^ containing 2% (*v*/*v*) heat-inactivated fetal bovine serum (2 × 10^6^ cells/mL), and chemotaxis was analyzed in 96-well ChemoTx chemotaxis chambers (Neuroprobe, Gaithersburg, MD). In brief, neutrophils were preincubated with the indicated concentrations of the test sample (essential oil or pure compound) or DMSO (1% final concentration) for 30 min at room temperature and added to the upper wells of the ChemoTx chemotaxis chambers. The lower wells were loaded with 30 µL of HBSS^+^ containing 2% (*v*/*v*) fetal bovine serum and the indicated concentrations of test sample, DMSO (negative control), or 1 nM *f*MLF as a positive control. Neutrophils were added to the upper wells and allowed to migrate through the 5.0 µm pore polycarbonate membrane filter for 60 min at 37 °C and 5% CO_2_. The number of migrated cells was determined by measuring ATP in lysates of transmigrated cells using a luminescence-based assay (CellTiter-Glo; Promega, Madison, WI), and luminescence measurements were converted to absolute cell numbers by comparison of the values with standard curves obtained with known numbers of neutrophils. Curve fitting (at least eight to nine points) and calculation of median effective concentration values (IC_50_) were performed by nonlinear regression analysis of the dose–response curves generated using GraphPad Prism 9.

### 2.10. Cytotoxicity Assay

Cytotoxicity of essential oils and pure compounds in human neutrophils or THP-1 monocytic was analyzed with a CellTiter-Glo Luminescent Cell Viability Assay Kit (Promega) according to the manufacturer’s protocol. Briefly, human neutrophils or THP-1 cells were cultured at a density of 10^4^ cells/well with different concentrations of essential oil or compound (final concentration of DMSO was 1%) for 90 min (for neutrophils) or 24 h (for THP-1 cells) at 37 °C and 5% CO_2_. Following treatment, substrate was added to the cells, and the samples were analyzed with a Fluoroscan Ascent FL microplate reader.

### 2.11. Physiochemical Properties of Compounds

The physicochemical properties of borneol were computed using SwissADME (http://www.swissadme.ch; accessed on 20 March 2022).

### 2.12. Statistical Analysis

One-way analysis of variance (ANOVA) was performed on the data sets, followed by Tukey’s pair-wise comparisons. Pair-wise comparisons with differences at *p* < 0.05 were considered statistically significant.

## 3. Results and Discussion

### 3.1. Composition of Essential Oils from G. squarrosa

Essential oils were extracted from *G.*
*squarrosa* flowers (designated as GEO_Fl_) and leaves (designated as GEO_Lv_). The distillation yield (*v*/*w*) for both essential oil samples was 0.5%. The chemical composition of these essential oils was evaluated using simultaneous GC-FID and GC/MS techniques. A total of 71 compounds, accounting for 97.4% and 96.2% of the flower and leaf essential oils, respectively, were identified and quantified (Table 1 and Table 2). The major classes of compounds in both essential oil samples were monoterpenes (89.7% and 88.6%) and sesquiterpenes (5.6% and 5.3%). The flower essential oils were enriched with oxygenated monoterpenes (46.5%), while the leaf essential oils were enriched in monoterpene hydrocarbons (47.2%). Major representatives of the monoterpene hydrocarbons (indicated as % GEO_Fl_/GEO_Lv_) were α-pinene (24.7/23.2), limonene (10.0/14.7), β-pinene (4.0/3.8), and terpinolene (1.7/2.0). The main oxygenated monoterpenes were (−)-borneol (23.4/16.6), *p*-cymen-8-ol (6.1/5.8), bornyl acetate (3.0/5.1), *trans*-pinocarveol (4.2/3.7), and myrtenol (2.5/1.7). The oxygenated sesquiterpene, spathulenol (3.0/2.0), was also present. Although the major components present in essential oils from *G.*
*squarrosa* flowers and leaves were similar, there were differences found between GEO_Fl_ and GEO_Lv_ for 12 of the minor components present (Table 1).

Comparison of our results with those reported previously for essential oils isolated from *G. squarrosa* collected in different countries showed that our samples generally had higher percentages of monoterpenes, namely α-pinene, β-pinene, limonene, and borneol. Essential oils from *G. squarrosa* grown in Romania were obtained from flowering shoots and had lower percentages of these monoterpenes, although they contained a higher amount of bornyl acetate (10.8%) [15,16]. Essential oils from *G. squarrosa* grown in Germany were obtained from flowers, leaves, and stems and contained similar levels of α-pinene (35.3, 10.4, and 4.4%), limonene (9.3, 16.2, and 27.1%), and β-pinene (5.2, 1.7, and 1.9%) but also had a unique, oxygenated monoterpene, (*iso*)-bornyl-C_5_-ester (6.9, 4.3, and 1.9%, respectively), and much higher levels of the sesquiterpene germacrene B (6.8, 13.2, and 3.0%, respectively) [13]. Notably, borneol was either not present or only present as a minor component in these other *G. squarrosa* essential oil samples [10,13,14]. Regarding other *Grindelia* species, essential oils isolated from *G. robusta* Nutt. grown in Germany were rich in oxygenated terpenes (57.4%), of which borneol (14.8%) was the largest constituent [10]. Essential oils from *G. humilis* Hook & Arn grown in Egypt had a completely different composition dominated by polyacetylenes (46.2%), such as (*E*)-lachnophyllol acetate, (2*E*,8*E*)-tridecadien-4,6-diyn-10-ol, 1-sencionyloxy-decadiyn-(4,6)-diene-(2*E*,8*E*), and their stereoisomers, as well a significant amount of the sesquiterpene hydrocarbon germacrene D (11.9%) [10]. The flower essential oils from *G. integrifolia* DC contained monoterpenes, such as α-pinene (34.9%) and limonene (13.1%), while the leaf essential oils contained myrcene (16.9%) and limonene (10.1%), as well as the sesquiterpenes spathulenol (12.3%) and β-eudesmol (11.9%) [12]. Finally, essential oils of *G. discoidea* Hook & Arn contained sesquiterpene alcohols, such as (*E,E*)-farnesol (˃9.0%) and (*E,Z*)-farnesol (15.7%) [11].

We report here the first enantioselective analysis of constituents in *G. squarrosa* essential oils, as all previous studies on *Grindelia* essential oils included identification of the essential oil constituents but not characterization of enantiomeric distribution. The enantiomeric distribution of monoterpenes in the flower and leaf essential oils of *G. squarrosa* was obtained using enantioselective GC analyses on chiral columns. As shown in Table 2, α-pinene, β-pinene, borneol, and camphor were present primarily or exclusively as (−)-enantiomers, while limonene was present primarily as the (+)-enantiomer. Differentiation of the enantiomeric composition of constituents is important for consideration of biological activity. Indeed, there are several examples of chiral compounds where the enantiomers had different immunomodulatory activity. For example, in vivo experiments showed that carvone enantiomers differentially modulated IgE-mediated airway inflammation in mice [52]. Likewise, the *R* isomer of hydroxychloroquine was found to exhibit higher antiviral activity and lower toxicity in vivo compared to the *S* isomer [53]. In additon, Murai et al. [54] used in silico analysis of enantioselective binding to characterize binding of immunomodulatory imide drugs to cereblon. Thus, it is clear that enantiomeric conformation of compounds has to be taken into consideration when evaluating biological effects.

### 3.2. Effect of the Essential Oils from G. squarrosa and (−)-Borneol on Neutrophil Ca^2+^ Influx

Neutrophils are the most abundant leukocyte and are vital for innate immunity [55]. These cells are the first responders to infection and injury in various tissues, establishing the first line of defense through multiple mechanisms such as phagocytosis, cytokine secretion, and reactive oxygen species production [56,57]. Thus, neutrophils represent an ideal pharmacological target for therapeutic development, and numerous natural products, including essential oils, have been shown to exhibit neutrophil immunomodulatory activity [29,30,31,32,33].

*Grindelia* essential oils were evaluated for their immunomodulatory effects on human neutrophils. Specifically, we evaluated their effects on [Ca^2+^]_i_, which is a key component of neutrophil activation and function [58]. We found that treatment of neutrophils with *Grindelia* essential oils activated human neutrophils, resulting in increased [Ca^2+^]_i_ (EC_50_ = 22.3 µg/mL for GEO_Fl_ and 19.4 µg/mL for GEO_Lv_). Previously, we found that several of the compounds that are also present in *G. squarrosa* essential oils (i.e., α-pinene, limonene, *p*-cymen-8-ol, β-pinene, terpinolene, spathulenol, and myrtenol) had no effect on human neutrophil Ca^2+^ influx [33,34], whereas (±)-bornyl acetate activated human neutrophils [59]. Analysis of the individual enantiomers (−)-borneol and (+)-borneol here showed that (−)-borneol activated human neutrophil [Ca^2+^]_i_ (EC_50_ = 28.7 ± 2.6), whereas (+)-borneol was inactive (Table 3), and representative kinetic curves for neutrophil [Ca^2+^]_i_ induced by (−)-borneol and (+)-borneol are shown in Figure 1. These results are consistent with our finding that only (−)-borneol was present in *G.*
*squarrosa* essential oils (Table 2), confirming that (−)-borneol is responsible for at least part of the neutrophil activation observed here for *G.*
*squarrosa* and suggesting that (−)-borneol may also be responsible for part of the neutrophil activation reported previously for *Artemisia*
*dracunculus* L. essential oils [59]. 

Since *Grindelia* essential oils and several component compounds stimulated human neutrophil [Ca^2+^]_i_, and it is well recognized that agonists can down-regulate neutrophil responses to subsequent treatment with heterologous or homologous agonists [60], we evaluated whether *Grindelia* essential oils and/or pure components could inhibit agonist-induced [Ca^2+^]_i_ in human neutrophils. As shown in Table 3 and Figure 2A, *Grindelia* essential oils inhibited [Ca^2+^]_i_ in *f*MLF- and WKYMVM-stimulated neutrophils with IC_50_ values in the micromolar range. We also evaluated the effect of (−)-borneol and (+)-borneol and found that only (−)-borneol inhibited *f*MLF- and WKYMVM-stimulated neutrophils (Table 3). A representative, concentration-dependent response for the inhibition of *f*MLF-induced neutrophil [Ca^2+^]_i_ by (−)-borneol is shown in Figure 2B. These results are also consistent with our data showing neutrophils were only activated by (−)-borneol (Table 3).

### 3.3. Effect of Essential Oils from G. squarrosa and Borneol on Neutrophil Chemotaxis 

Various essential oils and their components have been reported previously to inhibit neutrophil migration, including (±)-bornyl acetate, which is one of the major component compounds we found in *G.*
*squarrosa* essential oils [29,30,31,33,34]. In the present study, we evaluated effects of *G.*
*squarrosa* essential oils and pure (−)- or (+)-borneol on human neutrophil chemotaxis and found that pretreatment with GEO_Fl_ or GEO_Lv_ dose-dependently inhibited *f*MLF-induced human neutrophil chemotaxis (IC_50_ = 5.0 ± 1.6 and 4.1 ± 0.8 µg/mL, respectively) (Figure 3A). Likewise, pretreatment with (−)-borneol also inhibited *f*MLF-induced human neutrophil chemotaxis (IC_50_ = 5.8 ± 1.4 µM), whereas (+)-borneol had no effect (Figure 3B). 

To ensure that the effects of the essential oils from *G.*
*squarrosa* or (−)-borneol on neutrophil functional activity were not influenced by possible toxicity, we evaluated the cytotoxicity of the essential oil samples (up to 55 µg/mL) and (−)-borneol at various concentrations (up to 50 µM) in human neutrophils. We found that the *G.*
*squarrosa* essential oils were non-cytotoxic up to 55 µg/mL during a 90 min incubation period, which covers the times used to measure Ca^2+^ influx (up to 30 min) and cell migration (up to 90 min) (data not shown). Likewise, (−)-borneol was non-cytotoxic in human neutrophils treated for 90 min and THP-1 monocytic cells treated for 24 h (Figure 4), confirming previous reports on the low cytotoxicity of borneol [61]. Thus, we conclude that (−)-borneol is a novel, non-cytotoxic, innate immunomodulator.

In addition to the immunomodulatory activity reported here, borneol was previously reported to exhibit a number of biological activities. For example, borneol attenuated brain neuronal and microglial inflammation in lipopolysaccharide (LPS)-induced sepsis mice with suppression of p-p65 and p38 signaling that was initially activated by LPS in the brain [62]. Borneol treatment also inhibited transient receptor potential ankyrin 1 (TRPA1), a proinflammatory and noxious pain-sensing cation channel [63], and suppressed inflammatory responses in LPS-induced acute lung injury through inhibition of the nuclear factor κB (NF-κB) and mitogen-activated protein kinase (MAPK) signaling pathways [64]. Likewise, borneol-treated mice had reduced carrageenan-induced leukocyte migration to the peritoneal cavity [65], and borneol treatment suppressed proinflammatory cytokine mRNA expression in colonic inflammation [66]. Borneol attenuated asthma in mice by decreasing the CD4^+^ T cells’ infiltration [67]. Moreover, borneol specifically induced the activation of M2 macrophages in a signal transducer and activator of transcription 3 (STAT3)-dependent manner [68].

The biological activity of (−)-borneol was also reported previously. For example, (−)-borneol was recently reported to exhibit antibacterial activity [69]. (−)-Borneol was also reported to have neuroprotective effects in a middle cerebral artery occlusion (MCAO) model [70,71], as well as vasorelaxant properties, which may be attributed to Ca^2+^ influx blockade through voltage-gated Ca^2+^ channels, Ca^2+^ mobilization from intracellular stores, and activation of K^+^ channels [72,73,74]. Store-independent Ca^2+^ channels that exist in various hematopoietic cells, including neutrophils, are pharmacologically and/or immunologically similar to voltage-gated Ca^2+^ channels [75]. Thus, these store-independent Ca^2+^ channels could be potential targets of (−)-borneol in human neutrophils. Indeed, we showed previously that some essential oil compounds can modulate activity of transient receptor potential Ca^2+^ channels [33,76].

We also calculated the most important physicochemical parameters for borneol using SwissADME [77]. The logP values estimated using ALOGPS 2.1 program [78] and tPSA values allowed us to predict that borneol can permeate the blood–brain barrier (BBB) (Table 4). It should be noted that (−)-borneol itself can also increase blood–brain barrier permeability [73,79].

In previous research on essential oils from various plant species, we found that most essential oil compounds that inhibited *f*MLF-induced Ca^2+^ influx were sesquiterpenes, although one was an oxygenated monoterpene (bornyl acetate) (Table 5). In the present study, we showed that another oxygenated monoterpene, (−)-borneol, has neutrophil inhibitory activity. *Grindelia* essential oils also contained 2–3% spathulenol, an active, oxygenated sesquiterpene (Table 1 and Table 5). Thus, this compound may also be involved in the inhibitory effect of these essential oils, although we did not evaluate this compound here.

## 4. Conclusions

We analyzed the composition of essential oils extracted from *G. squarrosa* leaves and flowers and report here the first determination of the enantiomeric distribution of major monoterpenes in *G. squarrosa* essential oils. Borneol, α-pinene, β-pinene, and camphor were primarily present as (−)-enantiomers, whereas limonene was primarily present as the (+)-enantiomer. Further analysis of the immunomodulatory activity of *G. squarrosa* essential oils showed that they activated human neutrophils and were able to inhibit agonist-induced neutrophil activation and chemotaxis, which might contribute to the reported anti-inflammatory activity and other pharmacological properties of extracts from this plant. The effects of essential oils from *G.*
*squarrosa* might be attributable to (−)-borneol, bornyl acetate, some of the minor components, or synergetic effects among these constituents. However, to verify the key targets responsible for the immunomodulatory effects of (−)-borneol, further experimental investigation is needed.

## Figures and Tables

**Figure 1 molecules-27-04897-f001:**
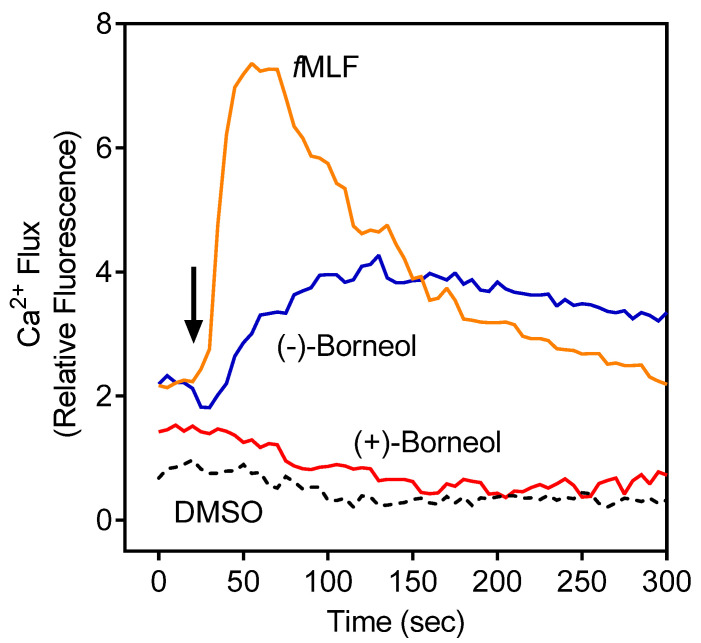
Direct effect of (−)-borneol and (+)-borneol on neutrophil [Ca^2+^]_i_. Human neutrophils were treated with 25 µM (−)-borneol, 25 µM (+)-borneol, 5 nM *f*MLF (positive control), or 1% DMSO (negative control), and [Ca^2+^]_i_ was monitored for the indicated times (arrow indicates when treatment was added). Data are from one experiment that is representative of three independent experiments.

**Figure 2 molecules-27-04897-f002:**
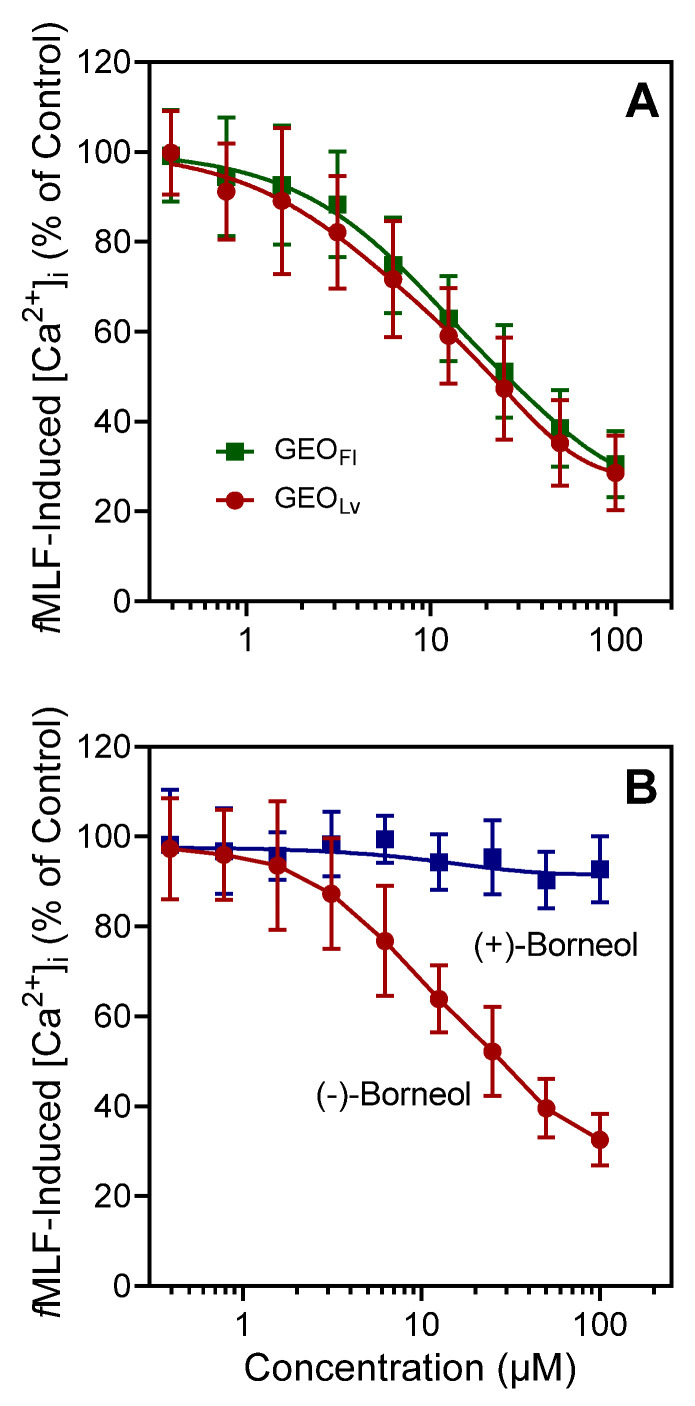
Effect of *Grindelia* essential oils or borneol on *f*MLF-induced neutrophil [Ca^2+^]_i_. Human neutrophils were treated with the indicated concentrations of GEO_Fl_ and GEO_Lv_ (**A**), (−)-borneol and (+)-borneol (**B**), or 1% DMSO (negative control) for 10 min. The cells were then activated by 5 nM *f*MLF, and [Ca^2+^]_i_ was monitored as described. The data shown are presented as the mean ± SD from one experiment that is representative of three independent experiments with similar results.

**Figure 3 molecules-27-04897-f003:**
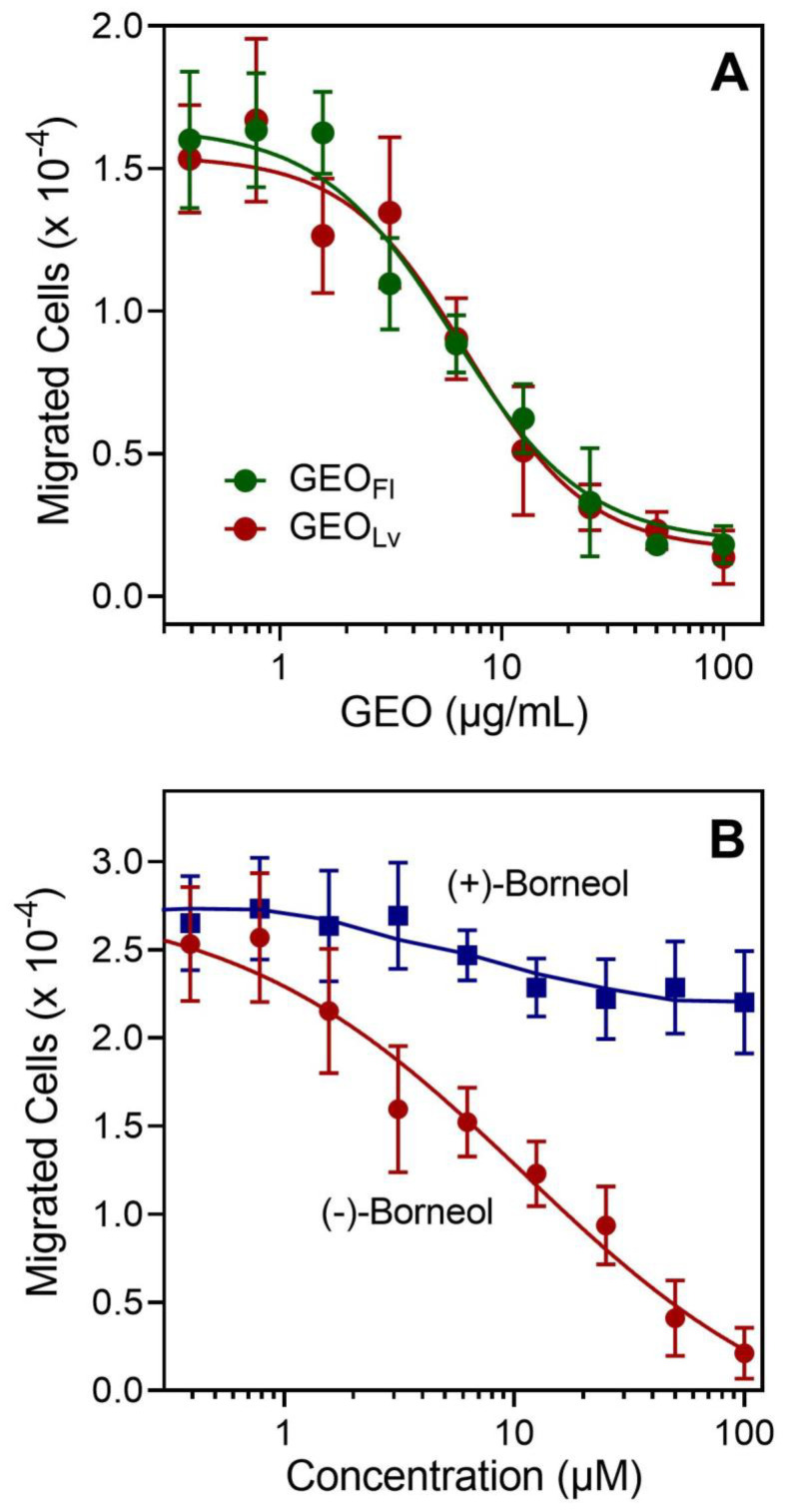
Effect of *Grindelia* essential oils or borneol on human neutrophil chemotaxis. Neutrophils were pretreated with the indicated concentrations of GEO_Fl_ and GEO_Lv_ (**A**) or (−)-borneol and (+)-borneol (**B**), and neutrophil migration toward 1 nM *f*MLF was measured, as described. The data are from one experiment that is representative of three independent experiments.

**Figure 4 molecules-27-04897-f004:**
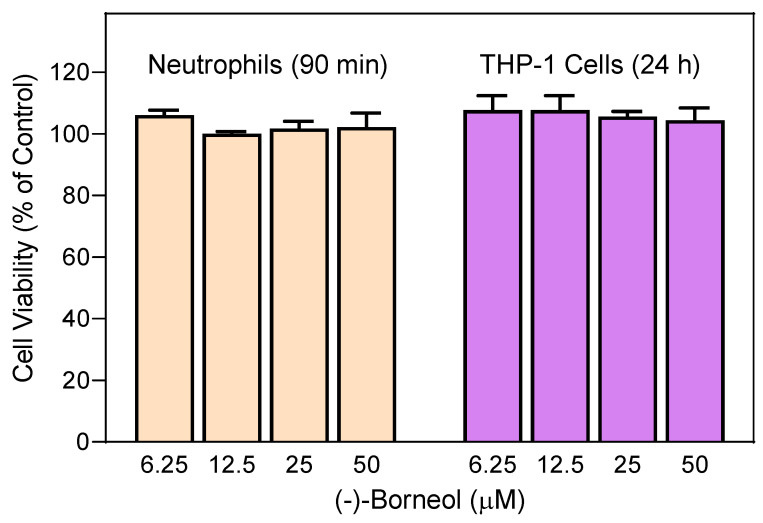
Cytotoxicity of (−)-borneol. Human neutrophils or human THP-1 monocytic cells were preincubated with indicated concentrations of (−)-borneol for 90 min or 24 h, and cell viability was analyzed, as described. Values are the mean ± SD of triplicate samples from one experiment that is representative of three independent experiments with similar results.

**Table 1 molecules-27-04897-t001:** Chemical composition of essential oils (%) isolated from flowers (GEO_Fl_) and leaves (GEO_Lv_) of *G. squarrosa*.

No	RRI	Pub RRI	Compound	GEO_Fl_	GEO_Lv_	No	RRI	Pub RRI	Compound	GEO_Fl_	GEO_Lv_
1	1032	1008–1039 ^a^	α-Pinene	24.7	23.2	37	1648	1597–1648 ^a^	Myrtenal	0.8	1.0
2	1076	1043–1086 ^a^	Camphene	1.2	1.5	38	1650	1612–1654 ^a^	δ-Elemene	t	0.1
3	1118	1085–1130 ^a^	β-Pinene	4.0	3.8	39	1670	1643–1671 ^a^	*trans*-Pinocarveol	4.2	3.7
4	1132	1098–1140 ^a^	Sabinene	t	0.1	40	1674	1670–1740 ^a^	*p*-Mentha-1,5-dien-8-ol	t	0.1
5	1150	1140 ^b^	Thuja-2,4(10)-diene	t	t	41	1683	1665–1691 ^a^	*trans*-Verbenol	1.9	2.0
6	1159	1122–1169 ^a^	δ-3-Carene		t	42	1700	1681 ^d^	*p*-Mentha-1,8-dien-4-ol	0.2	0.1
7	1174	1140–1175 ^a^	Myrcene	0.3	0.4	43	1706	1659–1724 ^a^	α-Terpineol	1.7	0.5
8	1176	1148–1186 ^a^	α-Phellandrene	t	t	44	1719	1653–1728 ^a^	Borneol	23.4	16.6
9	1188	1154–1195 ^a^	α-Terpinene	t	t	45	1726	1676–1726 ^a^	Germacrene D	0.2	0.3
10	1195	1167–1197 ^a^	Dehydro-1,8-cineole	t	t	46	1751	1699–1751 ^a^	Carvone		0.4
11	1203	1178–1219 ^a^	Limonene^d^	10.0	14.7	47	1763	1771 ^e^	Isobornyl isovalerate		0.3
12	1213	1186–1231 ^a^	1,8-Cineole	t	t	48	1773	1722–1774 ^a^	δ-Cadinene	0.3	
13	1218	1188–1233 ^a^	β-Phellandrene	0.2	0.2	49	1796	1750–1800 ^a^	Selina-3,7(11)-diene	0.3	0.3
14	1246	1211–1251 ^a^	(*Z*)-β-Ocimene	0.3	0.1	50	1797	1739–1797 ^a^	*p*-Methyl acetophenone	0.2	0.3
15	1255	1222–1266 ^a^	γ-Terpinene	0.1		51	1804	1743–1808 ^a^	Myrtenol	2.5	1.7
16	1266	1232–1267 ^a^	(*E*)-β-Ocimene	0.2	0.6	52	1854	1778–1854 ^a^	Germacrene B	0.4	0.5
17	1280	1246–1291 ^a^	*p*-Cymene	0.5	0.6	53	1864	1813–1865 ^a^	*p*-Cymen-8-ol	6.1	5.8
18	1290	1260–1300 ^a^	Terpinolene	1.7	2.0	54	1882	1818–1882 ^a^	*cis*-Carveol	t	
19	1328	1328 ^f^	2,2,6-Trimethylcyclohexanone		t	55	1949	1840–1949 ^a^	Piperitenone	t	
20	1384	1331–1384 ^a^	α-Pinene oxide		t	56	1992	1954–1992 ^a^	2-Phenylethyl-3-methylbutyrate	t	
21	1452	1452 ^g^	α,*p*-Dimethylstyrene		0.1	57	2088	2026–2090 ^a^	1-*epi*-Cubenol	t	t
22	1477	1477 ^c^	4,8-Epoxyterpinolene	0.3	0.5	58	2095	2033–2097 ^a^	Hexyl benzoate	t	
23	1494	1494 ^h^	(*Z*)-3-Hexenyl 3-methylbutyrate	0.2	t	59	2098	2049–2104 ^a^	Globulol	0.3	0.3
24	1495	1471–1495 ^a^	Bicycloelemene		0.1	60	2115	2115 ^i^	4-Hydroxy-4-methyl-cyclohex-2-enone	0.9	0.7
25	1499	1477–1511 ^a^	α-Campholene aldehyde		0.2	61	2144	2074–2150 ^a^	Spathulenol	3.0	2.0
26	1522	1482–1522 ^a^	Chrysanthenone		0.1	62	2164	2154 ^j^	Muurola-4,10(14)dien-1-ol		0.3
27	1532	1481–1537 ^a^	Camphor	0.5	0.9	63	2187	2135–2219 ^a^	τ-Cadinol		0.2
28	1549	1518–1560 ^a^	β-Cubebene	t		64	2247	2247 ^h^	*trans*-α-Bergamotol	0.2	0.3
29	1553	1507–1564 ^a^	Linalool	0.1	0.1	65	2255	2180–2255 ^a^	α-Cadinol	t	0.2
30	1562	1511–1562 ^a^	Isopinocamphone	0.7	1.6	66	2257	2196–2272 ^a^	β-Eudesmol	t	0.4
31	1571	1557–1625 ^a^	*trans-p*-Menth-2-en-1-ol	t		67	2260	2262 ^k^	Alismol	0.5	0.3
32	1586	1545–1590 ^a^	Pinocarvone	t		68	2300	2300	Tricosane	t	
33	1590	1549–1597 ^a^	Bornyl acetate	3.0	5.1	69	2308	2339 ^l^	13-epi-Manoyl oxide	0.8	0.9
34	1611	1564–1630 ^a^	Terpinen-4-ol	0.9	0.5	70	2320	-	Guaia-6,10(14)-dien-4-ol isomer ^#^	0.4	
35	1628	1583–1668 ^a^	Aromadendrene	t		71	2500	2500	Pentacosane		0.3
36	1639	1611–1688 ^a^	*trans-p*-Mentha-2,8-dien-1-ol	0.2	0.2						
**Summary of the Oil Composition**											
**Major Components**				**GEO_Fl_**					**GEO_Lv_**		
Monoterpene hydrocarbons				43.2					47.2		
Oxygenated monoterpenes				46.5					41.4		
Sesquiterpene hydrocarbons				1.2					1.3		
Oxygenated sesquiterpenes				4.4					4.0		
Miscellaneous compounds				2.1					2.3		
Total				97.4					96.2		

**Legend**: The data are presented as relative % for each component that was identified in the essential oils. %, calculated from flame ionization detector data. Trace amounts (t) were present at <0.1%. RRI, relative retention index calculated on the basis of retention of *n*-alkanes; Pub RRI, relative retention index published in ^a^ [41], ^b^ [42], ^c^ [43], ^d^ [44], ^e^ [45], ^f^ [46], ^g^ [47], ^h^ [48], ^i^ [31], ^j^ [49], ^k^ [50], and ^l^ [51]. ^#^ Tentatively identified using Wiley and MassFinder mass spectra libraries and published RRI. All other compounds were identified by comparison with co-injected standards.

**Table 2 molecules-27-04897-t002:** Enantiomeric distribution of *G. squarrosa* essential oils.

Compound		GEO_Fl_		GEO_Lv_	
		(%)	ee (%)	(%)	ee (%)
α-Pinene	(1*S*,5*S*)-(−)	100	100	100	100
	(1*R*,5*R*)-(+)	0		0	
β-Pinene	(1*S*,5*S*)-(−) (1*R*,5*R*)-(+)	91 9	82	89 11	78
Borneol	(1*S*,2*R*,4*S*)-(−)	100	100	100	100
	(1*R*,2*S*,4*R*)-(+)	0		0	
Camphor	(1*S*,4*S*)-(−)	95	90	97	94
	(1*R*,4*R*)-(+)	5		3	
Limonene	(4*S*)-(−)	3		2	
	(4*R*)-(+)	97	94	98	96

**Legend**: ee: enantiomeric excess.

**Table 3 molecules-27-04897-t003:** Effect of essential oils from *G.*
*squarrosa* and pure major component compounds on [Ca^2+^]_i_, chemotaxis, and cytotoxicity in human neutrophils.

Essential Oil	Activation of [Ca^2+^]_i_	Inhibition of [Ca^2+^]_i_	
		*f*MLF-Induced	WKYMVM-Induced
	EC_50_ (μg/mL)	IC_50_ (μg/mL)	
GEO_Fl_	22.3 ± 5.7	16.4 ± 3.9	1.6 ± 0.7
GEO_Lv_	19.4 ± 5.3	16.9 ± 2.0	3.4 ± 1.2
**Pure Compound**	**EC_50_ (μM)**	**IC_50_ (μM)**	
(−)-Borneol	28.7 ± 2.6	36.1 ± 1.5	54.2 ± 11.2
(+)-Borneol	N.A.	N.A.	N.A.
(±)-Bornyl acetate *	50.1 ± 11.5	42.6 ± 9.7	19.1 ± 0.1

**Legend**: EC_50_ and IC_50_ values were determined by nonlinear regression analysis of the dose–response curves as described under Section 2. For cytotoxicity study, human neutrophils were incubated with indicated concentrations of the compounds for 90 min, and cell viability was analyzed. N.A. indicates the samples had essentially no activity (EC_50_ or IC_50_ >55 µM for pure compounds or >55 µg/mL for the oils). The data are presented as the mean ± SD of three independent experiments. * reported in [59].

**Table 4 molecules-27-04897-t004:** Physicochemical properties of borneol, according to SwissADME results and binary classification tree model.

Property	Borneol
Formula	C_10_H_18_O
M.W.	154.25
Heavy atoms	11
Fraction Csp^3^	1.00
Rotatable bonds	0
H-bond acceptors	1
H-bond donors	1
MR	46.60
tPSA	20.23
LogP	2.83
BBB permeation	Yes

**Abbreviations**: M.W., molecular weight (g/mol); MR, molar refractivity; tPSA, topological polar surface area (Å^2^); LogP, lipophilicity; BBB, blood–brain barrier.

**Table 5 molecules-27-04897-t005:** Inhibitory effect of pure component compounds of essential oils on [Ca^2+^]_i_ in human neutrophils.

Compound	Chemical Class	IC_50_ (uM)	Reference
(−)-Borneol	Oxygenated monoterpene	36.1 ± 1.5	Present work
Bornyl Acetate	Oxygenated monoterpene	42.6 ± 9.7	[59]
Cedrol	Oxygenated sesquiterpene	15.4 ± 4.3	[29]
Curzerene	Oxygenated sesquiterpene	11.0 ± 3.8	[30]
Farnesene	Sesquiterpene hydrocarbone	1.1 ± 0.2	[59]
Germacrene D	Sesquiterpene hydrocarbone	0.5 ± 0.1	[31]
Germacrone	Oxygenated sesquiterpene	27.9 ± 8.9	[30]
6-Methyl-3,5-heptadien-2-one	Enone	8.2 ± 2.5	[34]
Spathulenol	Oxygenated sesquiterpene	36.2 ± 8.2	[30]
Viridiflorol	Oxygenated sesquiterpene	7.8 ± 2.3	[30]
Xanthoxylin	Alkyl-phenylketone	27.2 ± 6.6	[59]
α-Humulene	Sesquiterpene hydrocarbone	0.3 ± 0.1	[31]
β-Caryophyllene	Sesquiterpene hydrocarbone	0.33 ± 0.02	[31]

## Data Availability

Data are contained within the article and Appendix A.

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
