# Peer review of "Neutrophil Immunomodulatory Activity of (−)-Borneol, a Major Component of Essential Oils Extracted from *Grindelia squarrosa"

_molecules, 2022, doi:10.3390/molecules27154897_

Round 1

Reviewer 1 Report

Please see the manuscript

Author Response

Reviewer 1

Please see the manuscript.

 Answers:

  • We deleted the phrase on line 16 (“of the oils”).
  • We added a reference for the sentence on lines 257-262.
  • The typo on line 308 was corrected.
  • We added Artemisia dracunculus L. on line 319.
  • Table 4 was changed to Table 3.
  • Indicated issues for ref. 21 and 39 were corrected

Reviewer 2 Report

Dear Authors,

This study investigates the Neutrophil Immunomodulatory Activity of (−)-Borneol, a Major Component of Essential Oils Extracted from Grindelia squarrosa. The study is interesting, informative, and meaningful, especially the enantiomer analysis part.

Nevertheless, there are several issues that should be addressed before publication.

1

Abstract

Line 17-18: …..(−)-β-pinene (4.0 and 3.8%)

Herein, the percentage values for compounds with enantiomeric purity came from Table 1. However, the composition data in Table 1 were those for racemates without enantiomeric distribution information. Please check this.

Besides, it seems the enantiomeric distribution data obtained from chiral detection was based on MSD rather than FID (Line 137), which might be not reliable.

Therefore, I suggest the authors could state in the Abstract the composition data according to Table 1, and afterwards state the enantiomeric distribution information separately.

2

Table 1

Please check carefully the identification of compounds.

RRI values of experimental and literature must be both provided and compared in the table. There should be added a new column, for RRI literature.

3

Figure S3

The annotations of the peaks seem wrong. Please check and correct the mistakes.

The peaks of (−)-Camphor std and the essential oil may be mistakenly noted.

4

Figure S5
There should be three stacked chromatograms, (−)-Limonene std, (+)-Limonene std, and the essential oil.

The chromatogram of (−)-Limonene std was missing.

5

Others

Line 336: As shown in Table 4, Grindelia essential oils inhib….

The Table 4 did not contain the information on IC50 values (it is about the results from PharmMapper).

Please correct this mistake.

In addition, I suggest the authors should insert into the Figure 2 the Effect of Grindelia essential oil on fMLF-induced neutrophil [Ca2+]i, i.e. the variation with concentration. (can be presented as Figure 2A), to be compared directly with (-)-Borneol.

Also, please insert into Figure 3B the effects of pure (+)-borneol on human neutrophil chemotaxis, and compare the effects of (+)-borneol and (-)-borneol.

6

The molecular modeling part (PharmMapper) is useful, but it seems not associated so much with the above-mentioned experimental results obtained in this manuscript. Besides, the molecular modeling failed to give FPRs as the potential target for the component (-)-borneol.

Moreover, there is no experiment done on the effects of borneol on p38 MAPK, to validate the results from PharmMapper. Therefore, in my opinion, this part seems dispensable.

7

Did the authors consider other components, such as α-pinene, β-pinene, Limonene and camphor, for their potential biological activities towards human neutrophils? Sometimes the minor components might be also responsible for the activities of the EO.

8

The author team has published several works on the neutrophil immunomodulatory activity of essential oils. The authors could compare the activities of (-)-borneol in this study to those of other compounds (farnesene, etc.) investigated in previous studies, and discuss this in the discussion section, to emphasize the significance of the results on (-)-borneol in this study.

Farnesene:

Schepetkin, I.A.; Özek, G.; Özek, T.; Kirpotina, L.N.; Khlebnikov, A.I.; Klein, R.A.; Quinn, M.T. Neutrophil Immunomodulatory Activity of Farnesene, a Component of Artemisia dracunculus Essential Oils. Pharmaceuticals 202215, 642. https://doi.org/10.3390/ph15050642

Geranylacetone:

J. Agric. Food Chem. 2016, 64, 38, 7156–7170

(Z)-propenyl sec-butyl disulfide:

Oüzek, G., Schepetkin, I.A., Utegenova, G.A., Kirpotina, L.N., Andrei, S.R., Oüzek, T., Baser, K.H.C., Abidkulova, K.T., Kushnarenko, S.V., Khlebnikov, A.I., Damron, D.S. and Quinn, M.T. (2017), Chemical composition and phagocyte immunomodulatory activity of Ferula iliensis essential oils. Journal of Leukocyte Biology, 101: 1361-1371. https://doi.org/10.1189/jlb.3A1216-518RR

Thank you very much!

Author Response

Reviewer 2

This study investigates the Neutrophil Immunomodulatory Activity of (−)-Borneol, a Major Component of Essential Oils Extracted from Grindelia squarrosa. The study is interesting, informative, and meaningful, especially the enantiomer analysis part. Nevertheless, there are several issues that should be addressed before publication.

  1. Line 17-18: …..(−)-β-pinene (4.0 and 3.8%). Herein, the percentage values for compounds with enantiomeric purity came from Table 1. However, the composition data in Table 1 were those for racemates without enantiomeric distribution information. Please check this. Besides, it seems the enantiomeric distribution data obtained from chiral detection was based on MSD rather than FID (Line 137), which might be not reliable. Therefore, I suggest the authors could state in the Abstract the composition data according to Table 1, and afterwards state the enantiomeric distribution information separately.

Answer: The information about enantiomeric excess (ee) was added into the abstract. These data explain the enantiomeric ratio of each compound. All these percentages were calculated from FID detector integration, not from the mass detector. The information about the FID detector used for analyses was present in section 2.4. However, we added additional information about the system in section 2.5.

  1. Table 1. Please check carefully the identification of compounds. RRI values of experimental and literature must be both provided and compared in the table. There should be added a new column, for RRI literature.

Answer: Published RRI values from the literature were added into Table 1, and new references are included.

  1. Figure S3. The annotations of the peaks seem wrong. Please check and correct the mistakes. The peaks of (−)-Camphor std and the essential oil may be mistakenly noted.

Answer: The chromatograms were corrected. Our previous study data was used for noting the time of elution of (+)-Camphor std on this chiral column, so we clarified that the arrow indicates the location where (+)-Camphor standard eluted.

  1. Figure S5. There should be three stacked chromatograms, (−)-Limonene std, (+)-Limonene std, and the essential oil. The chromatogram of (−)-Limonene std was missing.

Answer: The chromatograms were corrected. Our previous studies data was used for noting the elution time of (−)-Limonene std on this chiral column, so we clarified that the arrow indicates the location where (-)-Limonene standard eluted.

  1. Line 336: As shown in Table 4, Grindelia essential oils inhib… The Table 4 did not contain the information on IC50 values (it is about the results from PharmMapper). Please correct this mistake.

Answer: The table name was corrected to Table 3.

  1. In addition, I suggest the authors should insert into the Figure 2 the Effect of Grindelia essential oil on fMLF-induced neutrophil [Ca2+]i, i.e. the variation with concentration. (can be presented as Figure 2A), to be compared directly with (-)-Borneol. Also, please insert into Figure 3B the effects of pure (+)-borneol on human neutrophil chemotaxis, and compare the effects of (+)-borneol and (-)-borneol.

Answer: We added Figure 2A, which shows effects of the essential oils on fMLF-induced neutrophil [Ca2+]i. In addition, effect of (+)-borneol on human neutrophil chemotaxis was added in Figure 3B.

  1. The molecular modeling part (PharmMapper) is useful, but it seems not associated so much with the above-mentioned experimental results obtained in this manuscript. Besides, the molecular modeling failed to give FPRs as the potential target for the component (-)-borneol. Moreover, there is no experiment done on the effects of borneol on p38 MAPK, to validate the results from PharmMapper. Therefore, in my opinion, this part seems dispensable.

Answer: We agree with the reviewer that PharmMapper results are dispensable and deleted these results together with methods from the manuscript.

  1. Did the authors consider other components, such as α-pinene, β-pinene, Limonene and camphor, for their potential biological activities towards human neutrophils? Sometimes the minor components might be also responsible for the activities of the EO. The author team has published several works on the neutrophil immunomodulatory activity of essential oils. The authors could compare the activities of (-)-borneol in this study to those of other compounds (farnesene, etc.) investigated in previous studies, and discuss this in the discussion section, to emphasize the significance of the results on (-)-borneol in this study.

Answer: We added a discussion of the various compounds we have identified and added a new Table (Table 5) to compare their activities.  In previous research on essential oils from various plant species, we found that most essential oil compounds that inhibited fMLF-induced Ca2+ influx were sesquiterpenes, although one was an oxygenated monoterpene (bornyl acetate). In the present studies, we show that another oxygenated monoterpene, (-)-borneol, has neutrophil inhibitory activity. Grindelia essential oils also contained 2-3% spathulenol, an active oxygenated sesquiterpene  Thus, this compound may also be involved in inhibitory effect of these essential oils.

Round 2

Reviewer 2 Report

Dear Authors,

Since all the issues have been addressed, in my opinion, the manuscript can be accepted for publication.

The study is interesting, informative, and meaningful. The authors are encouraged to perform further research on the specific mechanisms underlying the neutrophil immunomodulatory activity of the EOs and (−)-borneol in future.

Thank you!

Some minor errors to be corrected:

1

Abstract

Line 17:

‘in GEOLv and GEOFl,’

According to Table 1, it could be: in GEOFl and GEOLv (reversed).

Please check this.

2

Table 2

ee (%) for borenol is missing (100%).